# Identifying 1950s American Jazz Composers: Fine-Grained IsA Extraction via Modifier Composition

## Abstract

We present a method for populating fine-grained classes (e.g., *"1950s American jazz composers"*) with instances (e.g., *Charles Mingus*). While state-of-the-art methods tend to treat class labels as single lexical units, the proposed method considers each of the individual modifiers in the class label relative to the head. An evaluation on the task of reconstructing Wikipedia category pages demonstrates a >10 point increase in AUC, over a strong baseline relying on widely-used Hearst patterns.

## 1 Introduction

Substantial attention has been paid to automatically acquiring taxonomic knowledge, like that *"Charles Mingus"* is a *"composer"*, from text (Snow et al., 2006; Shwartz et al., 2016). The majority of approaches for extracting such "IsA" relations rely on lexical patterns as the primary signal of whether an instance belongs to a class: for example, observing a pattern like *"X such as Y"* is a strong indication that $Y$ is an instance of class $X$ (Hearst, 1992).

Methods based on these "Hearst patterns" assume that class labels can be treated as atomic lexicalized units. This assumption has several significant weakness. First, in order to recognize an instance of a class, these pattern-based methods require that the entire class label be observed verbatim in text. The requirement is reasonable for class labels containing a single word, but in practice, there are many possible fine-grained classes: not only *"composers"* but also *"1950s American jazz composers"*. The probability that a given label will appear in its entirety within one of the

| 1950s American jazz composers |
|---|
| …seminal **composers** *such as* **Charles Mingus** and George Russell… |
| …A virtuoso bassist and composer, **Mingus** irrevocably *changed the face of* **jazz**… |
| …**Mingus** truly *was a product of* America in all its historic complexities… |
| …**Mingus** *dominated the scene back in* the **1950s** and 1960s… |

Figure 1: We extract instances of fine-grained classes by considering each of the modifiers in the class label individually. This allows us to extract instances even when the full class label never appears in text.

expected patterns is very low, even in large amounts of text. Second, when class labels are treated as though they cannot be decomposed, every class label must be modeled independently, even those containing overlapping words (*"American jazz composer"*, *"French jazz composer"*). As a result, the number of meaning representations to be learned is exponential in the length of the class label, and quickly becomes intractable. Thus, compositional models of taxonomic relations are necessary for better language understanding.

We introduce a compositional approach for reasoning about fine-grained class labels. Our approach is based on the notion from formal semantics in which modifiers (*"1950s"*) correspond to properties which differentiate instances of a subclass (*"1950s composers"*) from instances of the superclass (*"composers"*) (Heim and Kratzer, 1998). Our method consists of two stages: interpreting each modifier relative to the head (*"composers active during 1950s"*), and using the interpretations to identify instances of the class from text (Figure 1). Our main contributions are: 1) a compositional method for IsA extraction, which in-

volves a novel application of noun-phrase para-phrasing methods to the task of semantic taxonomy induction and 2) the operationalization of a formal semantics framework to address two aspects of semantics that are often kept separate in NLP: assigning intrinsic "meaning" to a phrase, and reasoning about that phrase in a truth-theoretic context.

## 2 Related Work

**Noun Phrase Interpretation.** Compound noun phrases (*"jazz composer"*) communicate implicit semantic relations between modifiers and the head. Many efforts to provide semantic interpretations of such phrases rely on matching the compound to pre-defined patterns or semantic ontologies (Fares et al., 2015; Séaghdha and Copestake, 2007; Tratz and Hovy, 2010; Surtani and Paul, 2015; Choi et al., 2015). Recently, interpretations may take the form of arbitrary natural language predicates (Hendrickx et al., 2013). Most approaches are supervised, comparing unseen noun compounds to the most similar phrase seen in training (Wijaya and Gianfortoni, 2011; Nulty and Costello, 2013; Van de Cruys et al., 2013). Other unsupervised approaches apply information extraction techniques to paraphrase noun compounds (Kim and Nakov, 2011; Xavier and de Lima, 2014; Pasca, 2015). They focus exclusively on providing good paraphrases for an input noun compound. To our knowledge, ours is the first attempt to use these interpretations for the downstream task of IsA relation extraction.

**Semantic Taxonomy Induction.** Most efforts to learn taxonomic relations from text build on the seminal work of Hearst (1992), which observes that certain textual patterns–e.g., *"X and other Y"*–are high-precision indicators of whether $X$ is a member of class $Y$. Recent work focuses on learning such patterns automatically from corpora (Snow et al., 2006; Shwartz et al., 2016). These IsA extraction techniques provide a key step for the more general task of knowledge base population. The "universal schema" approach (Riedel et al., 2013; Kirschnick et al., 2016; Verga et al., 2016), which infers relations using matrix factorization, often includes Hearst patterns as input features. Graphical (Bansal et al., 2014)

and joint inference models (Movshovitz-Attias and Cohen, 2015) typically require Hearst patterns to define an inventory of possible classes. A separate line of work avoids Hearst patterns by instead exploiting semi-structured data from HTML markup (Wang and Cohen, 2009; Dalvi et al., 2012; Pasupat and Liang, 2014). These approaches all share the limitation that, in practice, in order for a class to be populated with instances, the entire class label has to have been observed verbatim in text. This requirement limits the ability to handle arbitrarily fine-grained classes. Our work addresses this limitation by modeling fine-grained class labels compositionally. Thus the proposed method can combine evidence from multiple sentences, and can perform IsA extraction without requiring any example instances of a given class.[1]

## 3 Modifiers as Functions

**Formalization.** In formal semantics, modification is modeled as function application. Specifically, let $MH$ be a class label consisting of a head $H$, which we assume to be a common noun, preceded by a modifier $M$. We use $[\![\cdot]\!]$ to represent the "interpretation function" which maps a linguistic expression to its denotation in the world. The interpretation of a common noun is the set of entities[2] in the universe $\mathcal{U}$ which are denoted by the noun (Heim and Kratzer, 1998):

$$[\![H]\!] = \{e \in \mathcal{U} \mid e \text{ is a } H\} \qquad (1)$$

The interpretation of a modifier $M$ is a function that maps between sets of entities. That is, modifiers select a subset[3] of the input set:

$$[\![M]\!](H) = \{e \in H \mid e \text{ satisfies } M\} \qquad (2)$$

This formalization leaves open how one decides whether or not "$e$ satisfies $M$". This non-trivial, as the meaning of a modifier can vary depending on the class it is modifying: if $e$ is a *"good student"*, $e$ is not necessarily a *"good*

---

[1]Pasupat and Liang (2014) also focuses on zero-shot IsA extraction, but exploits HTML document structure, rather than reasoning compositionally.

[2]We use "entities" and "instances" interchangeably; "entities" is standard terminology in linguistics.

[3]As does virtually all previous work in information extraction, we assume that modifiers are *subsective*, acknowledging the limitations (Kamp and Partee, 1995).

*person"*, making it difficult to model whether *"e satisfies good"* in general. We therefore reframe the above equation, so that the decision of whether *"e satisfies M"* is made by calling a binary function $\phi_M$, parameterized by the class $H$ within which $e$ is being considered:

$$\llbracket M \rrbracket(H) = \{e \in H \mid \phi_M(H, e)\} \qquad (3)$$

Conceptually, $\phi_M$ captures the core "meaning" of the modifier $M$, which is the set of properties that differentiate members of the output class $MH$ from members of the more general input class $H$. This formal semantics framework has two important consequences. First, the modifier has an intrinsic "meaning". The properties entailed by the modifier are independent of the particular state of the world. This makes it possible to make inferences about *"1950s composers"* even if no 1950s composers have been observed. Second, the modifier is a function that can be applied in a truth-theoretic setting. That is, applying *"1950s"* to the set of *"composers"* returns exactly the set of *"1950s composers"*.

**Computational Approaches.** While the notion of modifiers as functions has been incorporated into computational models previously, prior work focuses on either assigning an intrinsic meaning to $M$ or on operationalizing $M$ in a truth-theoretic sense, but not on doing both simultaneously. For example, Young et al. (2014) focuses exclusively on the subset selection aspect of modification. That is, given a set of instances $H$ and a modifier $M$, their method could return the subset $MH$. However, their method does not model the meaning of the modifier itself, so that, e.g., if there were no red cars in their model of the world, the phrase *"red cars"* would have no meaning. In contrast, Baroni and Zamparelli (2010) models the meaning of modifiers explicitly as functions which map between vector-space representations of nouns. However, their model focuses on similarity between class labels–e.g., to say that *"important routes"* is similar to *"major roads"*–and it is not obvious how the method could be operationalized in order to identify instances of those classes. A contribution of our work is to model the semantics of $M$ intrinsically, but in a way that permits application in the

model theoretic setting. We learn an explicit model of the "meaning" of a modifier $M$ relative to a head $H$, represented as a distribution over properties which differentiate the members of the class $MH$ from those of the class $H$. We then use this representation to identify the subset of instances of $H$ which constitute the subclass $MH$.

## 4 Learning Modifier Interpretations

### 4.1 Setup

For each modifier $M$, we would like to learn the function $\phi_M$ from Eq. 3. Doing so makes it possible, given $H$ and an instance $e \in H$, to decide whether $e$ has the properties required to be an instance of $MH$. In general, there is no systematic way to determine the implied relation between $M$ and $H$, as modifiers can arguably express any semantic relation, given the right context (Weiskopf, 2007). We therefore model the semantic relation between $M$ and $H$ as a distribution over properties which could potentially define the subclass $MH \subseteq H$. We will refer to this distribution as a "property profile" for $M$ relative to $H$. We make the assumption that relations between $M$ and $H$ that are discussed more often are more likely to capture the important properties of the subclass $MH$. This assumption is not perfect (Section 4.4) but has given good results for paraphrasing noun phrases (Nakov and Hearst, 2013; Pasca, 2015). Our method for learning property profiles is based on the unsupervised method proposed by Pasca (2015), which uses query logs as a source of common sense knowledge, and rewrites noun compounds by matching $MH$ (*"American composers"*) to queries of the form "$H(.*)M$" (*"composers from America"*).

### 4.2 Inputs

We assume two inputs: 1) an IsA repository, $\mathcal{O}$, containing $\langle e, C \rangle$ tuples where $C$ is a category and $e$ is an instance of $C$, and 2) a fact repository, $\mathcal{D}$, containing $\langle s, p, o, w \rangle$ tuples where $s$ and $o$ are noun phrases, $p$ is a predicate, and $w$ is a confidence that $p$ expresses a true relation between $s$ and $o$. Both $\mathcal{O}$ and $\mathcal{D}$ are extracted from a sample of around 1 billion Web documents in English. The supplementary material gives additional details.

| Good property profiles | | | | Bad property profiles | |
|---|---|---|---|---|---|
| rice dish | French violinist | Led Zeppelin song | still life painter | child actor | risk manager |
| * serve with rice | * live in France | Led Zeppelin write * | * known for still life | * have child | * take risk |
| * include rice | * born in France | Led Zeppelin play * | * paint still life | * expect child | * be at risk |
| * consist of rice | * speak French | Led Zeppelin have * | still life be by * | * play child | * be aware of risk |

Table 1: Example property profiles learned by observing predicates that relate instances of class $H$ to modifier $M$ ($I_2$). Results are similar when using the class label $H$ directly ($I_1$). We spell out inverted predicates (Section 4.2) so wildcards (*) may appear as subjects or objects.

We instantiate $\mathcal{O}$ with an IsA repository constructed by applying Hearst patterns to the Web documents. Instances are represented as automatically-disambiguated entity mentions[4] which, when possible, are resolved to Wikipedia pages. Classes are represented as (non-disambiguated) natural language strings. We instantiate $\mathcal{D}$ with a large repository of facts extracted using in-house implementations of ReVerb (Fader et al., 2011) and OLLIE (Mausam et al., 2012). The predicates are extracted as natural language strings. Subjects and objects may be either disambiguated entity references or natural language strings. Every tuple is included in both the forward and the reverse direction. E.g. $\langle jazz, perform\ at, venue \rangle$ also appears as $\langle venue, \leftarrow perform\ at, jazz \rangle$, where $\leftarrow$ is a special character signifying inverted predicates. These inverted predicates simplify the following definitions. In total, $\mathcal{O}$ contains 1.1M tuples and $\mathcal{D}$ contains 30M tuples.

### 4.3 Building Property Profiles

**Properties.** Let $I$ be a function which takes as input a noun phrase $MH$ and returns a property profile for $M$ relative to $H$. We define a "property" to be an SPO tuple in which the subject position[5] is a wildcard, e.g. $\langle *, born\ in, America \rangle$. Any instance which fills the wildcard slot then "has" the property. We expand adjectival modifiers to encompass nominalized forms using a nominalization dictionary extracted from WordNet (Miller, 1995). If $MH$ is *"American composer"* and we require a tuple to have the form $\langle H, p, M, w \rangle$, we will include tuples in which the third element is either *"American"* or *"America"*.

---

[4] "Entity mentions" may be individuals, like *"Barack Obama"*, but may also be concepts like *"jazz"*.

[5] Inverse predicates capture properties in which the wildcard is conceptually the object of the relation, but occupies the subject slot in the tuple. For example, $\langle venue, \leftarrow perform\ at, jazz \rangle$ captures that a *"jazz venue"* is a *"venue"* $e$ such that *"jazz performed at $e$"*.

**Relating $M$ to $H$ Directly.** We first build property profiles by taking the predicate and object from any tuple in $\mathcal{D}$ in which the subject is the head and the object is the modifier:

$$I_1(MH) = \{\langle\langle p, M\rangle, w\rangle \mid \langle H, p, M, w\rangle \in \mathcal{D}\} \quad (4)$$

**Relating $M$ to an Instance of $H$.** We also consider an extension in which, rather than requiring the subject to be the class label $H$, we require the subject to be an instance of $H$.

$$I_2(MH) = \{\langle\langle p, M\rangle, w\rangle \mid \langle e, H\rangle \in \mathcal{O} \\ \wedge \langle e, p, M, w\rangle \in \mathcal{D}\} \quad (5)$$

**Modifier Expansion.** In practice, when building property profiles, we do not require that the object of the fact tuple match the modifier exactly, as suggested in Eq. 4 and 5. Instead, we follow Pasca (2015) and take advantage of facts involving distributionally similar modifiers. Specifically, rather than looking only at tuples in $\mathcal{D}$ in which the object matches $M$, we consider all tuples, but discount the weight proportionally to the similarity between $M$ and the object of the tuple. Thus, $I_1$ is computed as below:

$$I_1(MH) = \{\langle\langle p, M\rangle, w \times sim(M, N)\rangle \\ \mid \langle H, p, N, w\rangle \in \mathcal{D}\} \quad (6)$$

where $sim(M, N)$ is the cosine similarity between $M$ and $N$. $I_2$ is computed analogously. We compute $sim$ using a vector space built from Web documents following Lin and Wu (2009); Pantel et al. (2009). We retain the 100 most similar phrases for each of ~10M phrases, and consider all other similarities to be 0.

### 4.4 Analysis of Property Profiles

Table 1 provides examples of good and bad property profiles for several $MH$s. In general, frequent relations between $M$ and $H$ capture relevant properties of $MH$, but it is not always

| Class label | Property profile |
|---|---|
| American company | * based in America |
| American composer | * born in America |
| American novel | * written in America |
| jazz album | * features jazz |
| jazz composer | * writes jazz |
| jazz venue | jazz performed at * |

Table 2: Head-specific property profiles learned by relating instances of $H$ to the modifier $M$ ($I_2$). Results are similar using $I_1$.

the case. To illustrate, the most frequently discussed relation between *"child"* and *"actor"* is that actors have children, but this property is not indicative of the meaning of *"child actor"*. Qualitatively, the top-ranked interpretations learned by using the head noun directly ($I_1$, Eq. 4) are very similar to those learned using instances of the head ($I_2$, Eq. 5). However, $I_2$ returns many more properties (10 on average per $MH$) than $I_1$ (just over 1 on average). Anecdotally, we see that $I_2$ captures more specific relations than does $I_1$. For example, for *"jazz composers"*, both methods return *"* write jazz"* and *"* compose jazz"*, but $I_2$ additionally returns properties like *"* be major creative influence in jazz"*. We compare $I_1$ and $I_2$ quantitatively in Section 6. Importantly, we do see that both $I_1$ and $I_2$ are capable of learning head-specific property profiles for a modifier. Table 2 provides examples.

## 5 Class-Instance Identification

**Instance finding.** After finding properties that relate a modifier to a head, we turn to the task of identifying instances of fine-grained classes. That is, for a given modifier $M$, we want to instantiate the function $\phi_M$ from Eq. 3. In practice, rather than being a binary function which decides whether or not $e$ is in class $MH$, our instantiation, $\hat{\phi}_M$, will return a real-valued score expressing the confidence that $e$ is a member of $MH$. For notational convenience, let $\mathcal{D}(\langle s, p, o \rangle) = w$, if $\langle s, p, o, w \rangle \in \mathcal{D}$ and 0 otherwise. We define $\hat{\phi}_M$ as follows:

$$\hat{\phi}_M(H, e) = \sum_{\langle \langle p, o \rangle, w \rangle \in I(MH)} w \times \mathcal{D}(\langle e, p, o \rangle) \tag{7}$$

Applying $M$ to $H$, then, is as in Eq. 3 except that instead of a discrete set, it returns a scored list of candidate instances:

$$[\![M]\!](H) = \{\langle e, \hat{\phi}_M(H, e) \rangle \mid \langle e, H \rangle \in \mathcal{O}\} \tag{8}$$

Ultimately, we need to identify instances of arbitrary class labels, which may contain multiple modifiers. Given a class label $C = M_1 \ldots M_k H$ which contains a head $H$ preceded by modifiers $M_1 \ldots M_k$, we generate a list of candidate instances by finding all instances of $H$ which have some property to support every modifier:

$$\bigcap_{i=1}^{k} \{\langle e, s(e) \rangle \mid \langle e, w \rangle \in [\![M_i]\!](H) \wedge w > 0\} \tag{9}$$

where $s(e)$ is the mean[6] of the scores assigned by each separate $\hat{\phi}_{M_i}$. From here on, we use **Mods** to refer to our method which generates lists of instances for a class using Eq. 8 and 9. When $\hat{\phi}_M$ (Eq. 7) is implemented using $I_1$, we use the name **Mods**$_H$ (for "heads"). When it is implemented using $I_2$, we use the name **Mods**$_I$ (for "instances").

**Weakly Supervised Reranking.** Eq. 8 uses a naive ranking in which the weight for $e \in MH$ is the product of how often $e$ has been observed with some property and the weight of that property for the class $MH$. Thus, instances of $H$ with overall higher counts in $\mathcal{D}$ receive high weights for every $MH$. We therefore train a simple logistic regression model to predict the likelihood that $e$ belongs to $MH$. We use a small set of features[7], including the raw weight as computed in Eq. 7. For training, we sample $\langle e, C \rangle$ pairs from our IsA repository $\mathcal{O}$ as positive examples and random pairs that were not extracted by any Hearst pattern as negative examples. We frame the task as a binary prediction of whether $e \in C$, and use the model's confidence as the value of $\hat{\phi}_M$ in place of the function in Eq. 7.

## 6 Evaluation

### 6.1 Experimental Setup

**Evaluation Sets.** We evaluate our models on their ability to return correct instances for arbitrary class labels. As a source of evaluation data, we use Wikipedia category pages[8]. These are pages in which the title is the name of the category (e.g., *"pakistani film actresses"*) and the body is a manually

---

[6] Also tried minimum, but mean gave better results.
[7] Feature templates in supplementary material.
[8] http://en.wikipedia.org/wiki/Help:Category

| |
|---|
| 2008 california wildfires · australian army chaplains · australian boy bands · canadian business journalists · canadian military nurses · canberra urban places · cellular automaton rules · chinese rice dishes · coldplay concert tours · daniel libeskind designs · economic stimulus programs · german film critics · invasive amphibian species · log flume rides · malayalam short stories · pakistani film actresses · puerto rican sculptors · string theory books · tampa bay devil rays scouts |

Table 3: Examples of labels from UniformSet.

curated list of links to other pages which fall under the category. We measure the precision and recall of each method for discovering the instances listed on these pages given the page title (henceforth "class label").

We collect the titles of all Wikipedia category pages, removing those in which the last word is capitalized or which contain fewer than three words. These heuristics are intended to retain compositional titles in which the head is a single common noun. We also remove any titles which contain links to sub-categories. This is to favor fine-grained classes (*"pakistani film actresses"*) over coarse-grained ones (*"film actresses"*). We perform heuristic modifier chunking in order to group together multiword modifiers (e.g., *"puerto rican"*); for details, see supplementary material. From the resulting list of class labels, we draw two samples of 100 labels each, enforcing that no $H$ appear as the head of more than three class labels per sample. The first sample is chosen uniformly at random (denoted **UniformSet**). The second (**WeightedSet**) is weighted so that the probability of drawing $M_1 \ldots M_k H$ is proportional to the total number of class labels in which $H$ appears as the head. Available at http://anonymized, these different evaluation sets are intended to evaluate performance on the head versus the tail of class label distribution, since information retrieval methods often perform differently on different parts of the distribution. On average, there are 17 instances per category in UniformSet and 19 in WeightedSet. Table 3 gives example class labels from UniformSet.

**Baselines.** We implement two baselines using our IsA repository ($\mathcal{O}$ as defined in Section 4.1). Our simplest baseline ignores modifiers altogether, and simply assumes that any instance of $H$ is an instance of $MH$, regardless of $M$. In this case the confidence value for $\langle e, MH \rangle$ is equivalent to that for $\langle e, H \rangle$. We

refer to this baseline simply as **Baseline**. Our second, stronger baseline uses the IsA repository directly to identify instances of the fine-grained class $C = M_1 \ldots M_k H$. That is, we consider $e$ to be an instance of the class if $\langle e, C \rangle \in \mathcal{O}$, meaning the entire class label appeared in a source sentence matching some Hearst pattern. We refer to this baseline as **Hearst**. The weight used to rank the candidate instances is the confidence value assigned by the Hearst pattern extraction (Section 4.2).

**Compositional Models.** As a baseline compositional model, we augment the Hearst baseline via set intersection. Specifically, for a class $C = M_1 \ldots M_k H$, if each of the $M_i H$ appears in $\mathcal{O}$ independently, we take the instances of $C$ to be the intersection of the instances of each of the $M_i H$. We assign the weight of an instance $e$ to be the sum of the weights associated with each independent modifier. We refer to this method as **Hearst∩**. We contrast this with our proposed model which recognizes instances of a fine-grained class by 1) assigning a meaning to each modifier in the form of a property profile and 2) checking whether a candidate instance exhibits these properties. We refer to the versions of our method as **Mods$_H$** and **Mods$_I$**, as described in Section 5. When relevant, we use "raw" to refer to the version in which instances are ranked using raw weights and "RR" to refer to the version in which instances are ranked using logistic regression (Section 5). We also try using the proposed methods to extend rather than replace the Hearst baseline. We combine predictions by merging the ranked lists produced by each system: i.e. the score of an instance is the inverse of the sum of its ranks in each of the input lists. If an instance does not appear at all in an input list, its rank in that list is set to a large constant value. We refer to these combination systems as **Hearst+Mods$_H$** and **Hearst+Mods$_I$**.

## 6.2 Results

**Precision and Coverage.** We first compare the methods in terms of their coverage, the number of class labels for which the method is able to find some instance, and their precision, to what extent the method is able to correctly rank true instances of the class above

| | |
|---|---|
| **Flemish still life painters:** Clara Peeters · Willem Kalf · Jan Davidsz de Heem · *Pieter Claesz* · ~~Peter Paul Rubens~~ · Frans Snyders · Jan Brueghel the Elder · ~~Hans Memling~~ · ~~Pieter Bruegel the Elder~~ · ~~Caravaggio~~ · Abraham Brueghel | |
| **Pakistani cricket captains:** Salman Butt · Shahid Afridi · Javed Miandad · Azhar Ali · ~~Greg Chappell~~ · Younis Khan · Wasim Akram · Imran Khan · Mohammad Hafeez · Rameez Raja · Abdul Hafeez Kardar · Waqar Younis · Sarfraz Ahmed | |
| **Thai buddhist temples:** *Wat Buddhapadipa* · Wat Chayamangkalaram · *Wat Mongkolratanaram* · ~~Angkor Wat~~ · ~~Preah Vihear Temple~~ · Wat Phra Kaew · Wat Rong Khun · Wat Mahathat Yuwaratrangsarit · ~~Vat Phou~~ · Tiger Temple · Sanctuary of Truth · Wat Chalong · ~~Swayambhunath~~ · ~~Mahabodhi Temple~~ · Tiger Cave Temple · Harmandir Sahib | |

Table 4: Instances extracted for several fine-grained classes from Wikipedia. Lists shown are from $\text{Mods}_I$. Instances in italics were also returned by Hearst∩. Strikethrough denotes incorrect.

non-instances. We report total coverage, the number of labels for which the method returns any instance, and correct coverage, the number of labels for which the method returns a correct instance. For precision, we compute the average precision (AP) for each class label. AP ranges from 0 to 1, where 1 indicates that all positive instances were ranked above all negative instances. We report mean average precision (MAP), which is the mean of the APs across all the class labels. MAP is only computed over class labels for which the method returns something, meaning methods are not punished for returning empty lists.

|  | UniformSet | | WeightedSet | |
|---|---|---|---|---|
|  | Coverage | MAP | Coverage | MAP |
| Baseline | 95 / 70 | 0.01 | 98 / 74 | 0.01 |
| Hearst | 9 / 9 | 0.63 | 8 / 8 | 0.80 |
| Hearst∩ | 13 / 12 | 0.62 | 9 / 9 | 0.80 |
| $\text{Mods}_H$ raw | 56 / 32 | 0.23 | 50 / 30 | 0.16 |
| $\text{Mods}_H$ RR | 56 / 32 | 0.29 | 50 / 30 | 0.25 |
| $\text{Mods}_I$ raw | 62 / 36 | 0.18 | 59 / 38 | 0.20 |
| $\text{Mods}_I$ RR | 62 / 36 | 0.24 | 59 / 38 | 0.23 |

Table 5: Coverage and precision for populating Wikipedia category pages with instances. "Coverage" is the number of class labels (out of 100) for which at least one instance was returned, followed by the number for which at least one correct instance was returned. "MAP" is mean average precision. MAP does not punish methods for returning empty lists, thus favoring the baseline (see Figure 2).

Table 4 gives examples of instances returned for several class labels and Table 5 shows the precision and coverage for each of the methods. Figure 2 illustrates how the single mean AP score (as reported in Table 5) can misrepresent the relative precision of different methods. In combination, Table 5 and Figure 2 demonstrate that the proposed methods extract instances about as well as the baseline, whenever the baseline can extract anything at all; i.e. the proposed method does not cause a precision drop on classes covered by the base-

line. In addition, there are many classes for which the baseline is not able to extract any instances, but the proposed method is.

Table 5 also reveals that the reranking model (RR) consistently increases MAP for the proposed methods. Therefore, going forward, we only report results using the reranking model (i.e. $\text{Mods}_H$ and $\text{Mods}_I$ will refer to $\text{Mods}_H$ RR and $\text{Mods}_I$ RR, respectively).

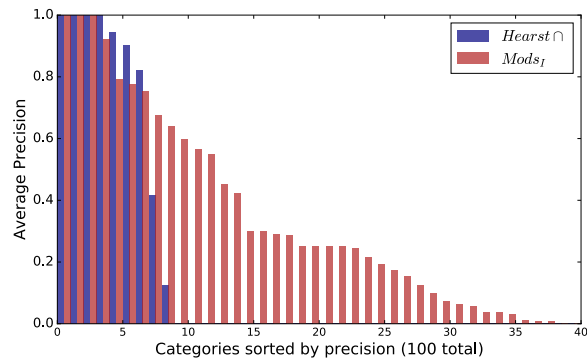

Figure 2: Distribution of AP over 100 class labels in WeightedSet. The proposed method (red) and the baseline method (blue) achieve high AP for the same number of classes, but $\text{Mods}_I$ additionally finds instances for classes for which the baseline returns nothing.

**Manual Re-Annotation.** It possible that true instances of a class are missing from our Wikipedia reference set, and thus that our precision scores underestimate the actual precision of the systems. We therefore manually verify the top 10 predictions of each of the systems for a random sample of 25 class labels. We choose class labels for which Hearst was able to return at least one instance, in order to ensure reliable precision estimates. For each of these labels, we manually check the top 10 instances proposed by each method to determine whether each belongs to the class. Table 6 shows the precision scores for each method computed against the

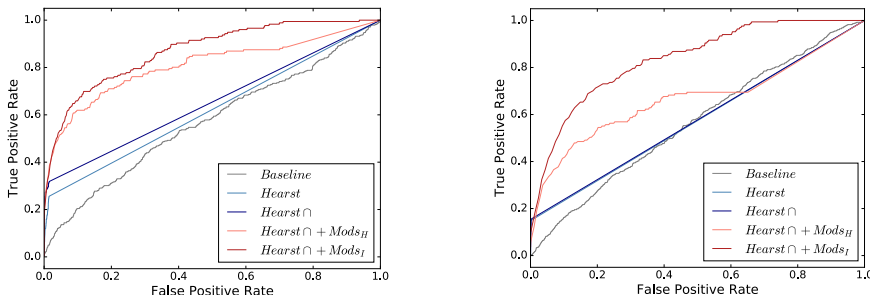

(a) Uniform random sample (UniformSet). (b) Weighted random sample (WeightedSet).

Figure 3: ROC curves for selected methods (Hearst in blue, proposed in red). Given a ranked list of instances, ROC curves plot true positives vs. false positives retained by setting various cutoffs. The curve becomes linear once all remaining instances have the same score (e.g., 0), as this makes it impossible to add true positives without also including all remaining false positives.

original Wikipedia list of instances and against our manually-augmented list of gold instances. The overall ordering of the systems does not change, but the precision scores increase notably after re-annotation. We continue to evaluate against the Wikipedia lists, but acknowledge that reported precision is likely an underestimate of true precision.

|  | Wikipedia | Gold |
|---|---|---|
| Hearst | 0.56 | 0.79 |
| Hearst∩ | 0.53 | 0.78 |
| $Mods_H$ | 0.23 | 0.39 |
| $Mods_I$ | 0.24 | 0.42 |
| Hearst+$Mods_H$ | 0.43 | 0.63 |
| Hearst+$Mods_I$ | 0.43 | 0.63 |

Table 6: P@10 before/after re-annotation; Wikipedia underestimates true precision.

**Precision-Recall Analysis.** We next look at the precision-recall tradeoff in terms of the area under the curve (AUC) achieved when each method attempts to rank the complete list of candidate instances. We take the union of all of the instances proposed by all of the methods (including the Baseline method which, given a class label $M_0 \ldots M_k H$, proposes every instance of the head $H$ as a candidate). Then, for each method, we rank this full set of candidates such that any instance returned by the method is given the score the method assigns, and every other instance is scored as 0. Table 7 reports the AUC and recall and Figure 3 plots the full ROC curves. The requirement by Hearst that class labels appear in full in a single sentence results in very low recall, which translates into very low AUC when considering the full set of candi-

date instances. By comparison, the proposed compositional methods make use of a larger set of sentences, and provide non-zero scores for many more candidates, resulting in a >10 point increase in AUC on both UniformSet and WeightedSet (Table 7).

|  | UniformSet | | WeightedSet | |
|---|---|---|---|---|
|  | AUC | Recall | AUC | Recall |
| Baseline | 0.55 | 0.23 | 0.53 | 0.28 |
| Hearst | 0.56 | 0.03 | 0.52 | 0.02 |
| Hearst∩ | 0.57 | 0.04 | 0.53 | 0.02 |
| $Mods_H$ | 0.68 | 0.08 | 0.60 | 0.06 |
| $Mods_I$ | 0.71 | 0.09 | 0.65 | 0.09 |
| Hearst∩+$Mods_H$ | 0.70 | 0.09 | 0.61 | 0.08 |
| Hearst∩+$Mods_I$ | 0.73 | 0.10 | 0.66 | 0.10 |

Table 7: Recall of instances on Wikipedia category pages, measured against the full set of instances from all pages in sample. AUC captures tradeoff between true and false positives.

# 7 Conclusion

We have presented an approach to IsA extraction which takes advantage of the compositionality of natural language. Existing approaches often treat class labels as atomic units which must be observed in full in order to be populated with instances. As a result, current methods are not able to handle the infinite number of classes describable in natural language, most of which never appear in text. Our method reasons about each modifier in the label individually, in terms of the properties that it implies about the instances. This approach allows us to harness information that is spread across multiple sentences, significantly increasing in the number of fine-grained classes which we are able to populate.

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
