# Peer review of "Identifying 1950s American Jazz Musicians: Fine-Grained IsA Extraction via Modifier Composition"

_ACL 2017 — decision unknown_

[Official Review · Reviewer 1 · rating 3 · confidence 5]
soundness 5 · originality 5 · clarity 4 · impact 3 · substance 2 · appropriateness 2 · meaningful comparison 3 · presentation format Poster

- Strengths:
 - the model if theoretically solid and motivated by formal semantics. 

- Weaknesses:

 - The paper is about is-a relation extraction but the majority of literature
about taxonomization is not referenced in the paper, inter alia:

Flati Tiziano, Vannella Daniele, Pasini Tommaso, Navigli Roberto.
2016. MultiWiBi: The multilingual Wikipedia bitaxonomy project.

Soren Auer, Christian Bizer, Georgi Kobilarov, Jens ¨
Lehmann, Richard Cyganiak, and Zachary Ive.
2007. DBpedia: A nucleus for a web of open data.

Gerard de Melo and Gerhard Weikum. 2010. MENTA:
Inducing Multilingual Taxonomies from Wikipedia.

Zornitsa Kozareva and Eduard H. Hovy. 2010. A
Semi-Supervised Method to Learn and Construct
Taxonomies Using the Web. 

Vivi Nastase, Michael Strube, Benjamin Boerschinger,
Caecilia Zirn, and Anas Elghafari. 2010. WikiNet:
A Very Large Scale Multi-Lingual Concept Network.

Simone Paolo Ponzetto and Michael Strube. 2007.
Deriving a large scale taxonomy from Wikipedia.

Simone Paolo Ponzetto and Michael Strube. 2011.
Taxonomy induction based on a collaboratively built
knowledge repository. 

Fabian M. Suchanek, Gjergji Kasneci, and Gerhard
Weikum. 2008. YAGO: A large ontology from
Wikipedia and WordNet. 

Paola Velardi, Stefano Faralli, and Roberto Navigli.
2013. OntoLearn Reloaded: A graph-based algorithm
for taxonomy induction. 

 - Experiments are poor, they only compare against "Hearst patterns" without
taking into account the works previously cited.

- General Discussion:
 The paper is easy to follow and the supplementary material is also well
written and useful, however the paper lack of references of is a relation
extraction and taxonomization literature. The same apply for the experiments.
In fact no meaningful comparison is performed and the authors not even take
into account the existence of other systems (more recent than hearst patterns).

I read authors answers but still i'm not convinced that they couldn't perform
more evaluations. I understand that they have a solid theoretical motivation
but still, i think that comparison are very important to asses if the
theoretical intuitions of the authors are confirmed also in practice. While
it's true that all the works i suggested as comparison build taxonomies, is
also true that a comparison is possible considering the edges of a taxonomy.

Anyway, considering the detailed author answer and the discussion with the
other reviewer i can rise my score to 3 even if i still think that this paper
is poor of experiments and does not frame correctly in the is-a relation
extraction / taxonomy building literature.

[Official Review · Reviewer 2 · rating 3 · confidence 2]
soundness 5 · originality 5 · clarity 3 · impact 3 · substance 3 · appropriateness 5 · meaningful comparison 3 · presentation format Poster

- Strengths:

This paper presents an approach for fine-grained IsA extraction by learning
modifier interpretations. The motivation of the paper is easy to understand and
this is an interesting task. In addition, the approach seems solid in general
and the experimental results show that the approach increases in the number of
fine-grained classes that can be populated.

- Weaknesses:

Some parts of the paper are hard to follow. It is unclear to me why D((e, p,
o)) is multiplied by w in Eq (7) and why the weight for e in Eq. (8) is
explained as the product of how often e has been observed with some property
and the weight of that property for the class MH. In addition, it also seems
unclear how effective introducing compositional models itself is in increasing
the coverage. I think one of the major factors of the increase of the coverage
is the modifier expansion, which seems to also be applicable to the baseline
'Hearst'. It would be interesting to see the scores 'Hearst' with modifier
expansion.

- General Discussion:

Overall, the task is interesting and the approach is generally solid. However,
since this paper has weaknesses described above, I'm ambivalent about this
paper.

- Minor comment:

I'm confused with some notations. For example, it is unclear for me what 'H'
stands for. It seems that 'H' sometimes represents a class such as in (e, H)
(- O, but sometimes represents a noun phrase such as in (H, p, N, w) (- D. Is
my
understanding correct?

In Paragraph "Precision-Recall Analysis", why the authors use area under the
ROC curve instead of area under the Precision-Recall curve, despite the
paragraph title "Precision-Recall Analysis"?

- After reading the response:

Thank you for the response. I'm not fully satisfied with the response as to the
modifier expansion. I do not think the modifier expansion can be applied to
Hearst as to the proposed method. However, I'm wondering whether there is no
way to take into account the similar modifiers to improve the coverage of
Hearst. I'm actually between 3 and 4, but since it seems still unclear how
effective introducing compositional models itself is, I keep my recommendation
as it is.

[Official Review · Reviewer 3 · rating 4 · confidence 3]
soundness 5 · originality 5 · clarity 5 · impact 3 · substance 4 · appropriateness 5 · meaningful comparison 3 · presentation format Poster

- strengths
This is a novel approach to modeling the compositional structure of complex
categories that maintains a set theoretic interpretation of common nouns and
modifiers, while also permitting a distributional interpretation of head
modification. The approach is well motivated and clearly defined and the
experiments show that show that this decomposed representation can improve upon
the Hearst-pattern derived IsA relations upon which it is trained in terms of
coverage.

- weaknesses
The experiments are encouraging. However, it would be nice to see ROC curves
for the new approach alone, not in an ensemble with Hearst patterns. Table 5
tells us that Mods_I increases coverage at the cost of precision and Figure 2
tells us that Mods_I matches Hearst pattern precision for the high precision
region of the data. However, neither of these tell us whether the model can
distinguish between the high and low precision regions, and the ROC curves
(which would tell us this) are only available for ensembled models.

I believe that Eqn. 7 has an unnecessary $w$ since it is already the case that
$w=D(\rangle e, p, o \langle)$.

- discussion
Overall, this is a nice idea that is well described and evaluated. I think this
paper would be a good addition to ACL.